# Post-concussion symptom burden and dynamics: Insights from a digital health intervention and machine learning

Rebecca Blundell[1], Christine d'Offay[2], Charles Hand[2], Daniel Tadmor[3], Alan Carson[4], David Gillespie[4], Matthew Reed[5,6], Aimun A. B. Jamjoom[2,7,8]*

1 Department of Neurosurgery, The National Hospital for Neurology and Neurosurgery, London, United Kingdom, 2 HeadOn Health Ltd, Edinburgh, United Kingdom, 3 Carnegie School of Sport, Leeds Beckett University, Leeds, United Kingdom, 4 Department of Clinical Neurosciences (DCN), Royal Infirmary of Edinburgh, Edinburgh, United Kingdom, 5 The Emergency Medicine Research Group Edinburgh (EMERGE), Royal Infirmary of Edinburgh, Edinburgh, United Kingdom, 6 Acute Care Edinburgh, Usher Institute, The University of Edinburgh, Edinburgh, United Kingdom Department of Clinical Neuroscience, Edinburgh Royal Infirmary, Edinburgh, United Kingdom, 7 Centre for Clinical Brain Sciences, The University of Edinburgh, Edinburgh, United Kingdom, 8 Department of Neurosurgery, Queen's Hospital, Romford, United Kingdom

* v1ajamjo@ed.ac.uk

**Data Availability Statement:** Data is available via Edinburgh DataShare (https://doi.org/10.7488/ds/7849).

## Abstract

Individuals who sustain a concussion can experience a range of symptoms which can significantly impact their quality of life and functional outcome. This study aims to understand the nature and recovery trajectories of post-concussion symptomatology by applying an unsupervised machine learning approach to data captured from a digital health intervention (HeadOn). As part of the 35-day program, patients complete a daily symptom diary which rates 8 post-concussion symptoms. Symptom data were analysed using K-means clustering to categorize patients based on their symptom profiles. During the study period, a total of 758 symptom diaries were completed by 84 patients, equating to 6064 individual symptom ratings. Fatigue, sleep disturbance and difficulty concentrating were the most prevalent symptoms reported. A decline in symptom burden was observed over the 35-day period, with physical and emotional symptoms showing early rates of recovery. In a correlation matrix, there were strong positive correlations between low mood and irritability (r = 0.84), and poor memory and difficulty concentrating (r = 0.83). K-means cluster analysis identified three distinct patient clusters based on symptom severity. Cluster 0 (n = 24) had a low symptom burden profile across all the post-concussion symptoms. Cluster 1 (n = 35) had moderate symptom burden but with pronounced fatigue. Cluster 2 (n = 25) had a high symptom burden profile across all the post-concussion symptoms. Reflecting the severity of the clusters, there was a significant relationship between the symptom clusters for both the Rivermead (p = 0.05) and PHQ-9 (p = 0.003) questionnaires at 6-weeks follow-up. By leveraging digital ecological momentary assessments, a rich dataset of daily symptom ratings was captured allowing for the identification of symptom severity clusters. These findings underscore the potential of digital technology and machine learning to enhance our understanding of post-concussion symptomatology and offer a scalable solution to support patients with their recovery.

**Funding:** This study was funded by The National Institute of Health and Care Research Brain Injury MedTech Co-operative (www.brainmic.nihr.ac.uk), Grant number R46241. It was also by a business grant from the Scottish EDGE (www.scottishedge.com). The funders did not play any role in study design, data collection, analysis or manuscript preparation/publication.

**Competing interests:** AJ, CD, DG and CH are shareholders in HeadOn Health Ltd, which has an exclusive license to commercialize the HeadOn intellectual property.

## Author summary

Individuals who sustain a concussion can experience a range of post-concussion symptoms that can significantly impact their quality of life. In this study, we used digital technology to capture daily symptom ratings from 84 patients over a 35-day period as they recovered from a concussion. This provided a rich dataset of over 6,000 individual symptom assessments, one of the largest collections of post-concussion symptom data to date. We found that fatigue, sleep disturbance, and difficulty concentrating were among the most prevalent and severe symptoms reported by patients. While the overall symptom burden declined over the 35 days, there was variation in how quickly different symptoms improved, with physical and emotional symptoms showing early recovery. By applying machine learning techniques, we were able to identify three distinct clusters of patients based on their symptom severity profile. These symptom clusters correlated with outcomes like depression and concussion symptomatology scores at 6 weeks. Our findings underscore how digital tools and data science approaches can provide novel insights into post-concussion symptomatology. Capturing granular, daily symptom data allowed us to characterize patient trajectories and identify differing symptom burden phenotypes. This scalable approach could help guide more personalized management strategies for patients recovering from concussion.

## Introduction

Individuals who sustain a concussion can experience a constellation of post-concussion symptoms, including physical (e.g., headaches and dizziness), cognitive (e.g., difficulty concentrating and memory problems), emotional (e.g., depression and anxiety) symptoms and sleep disturbance (hypo- or hyper-somnolence) [1]. Post-concussion symptomatology is highly heterogenous with a variety of symptom profiles and recovery trajectories [2]. Although symptom burden tends to improve over time, up to 40% or individuals can develop persistent symptoms up to a year after injury [3,4]. A high post-concussion symptom burden is associated with work absenteeism [5,6], poor quality of life [7] and difficulties with community integration [8]. Understanding the nature and recovery trajectories of post-concussion symptomatology is important to inform discussions with patients, guide management plans and allocate healthcare resources cost-effectively. Studies examining post-concussion symptoms have typically captured data at a limited number of time-points, usually weeks or months apart. We therefore lack a granular understanding of daily post-concussion symptom burden and dynamics. Scalable digital technology provides the opportunity to capture a large volume of symptom and recovery data from patients through digital ecological momentary assessments. We developed HeadOn, a digital health intervention designed to support patients with their recovery after a concussion [9]. During HeadOn patients share data on their symptoms and recovery. In this article, we describe the analysis of digitally collected data using machine learning approaches with the aim of providing novel insights into post-concussion symptom clusters and recovery trajectories.

## Materials and methods

### Digital Health Intervention

HeadOn is a digital health intervention designed to support patients with their recovery after a concussion (**Fig 1**). It was developed using a systematic evidence-, theory-, and person-based

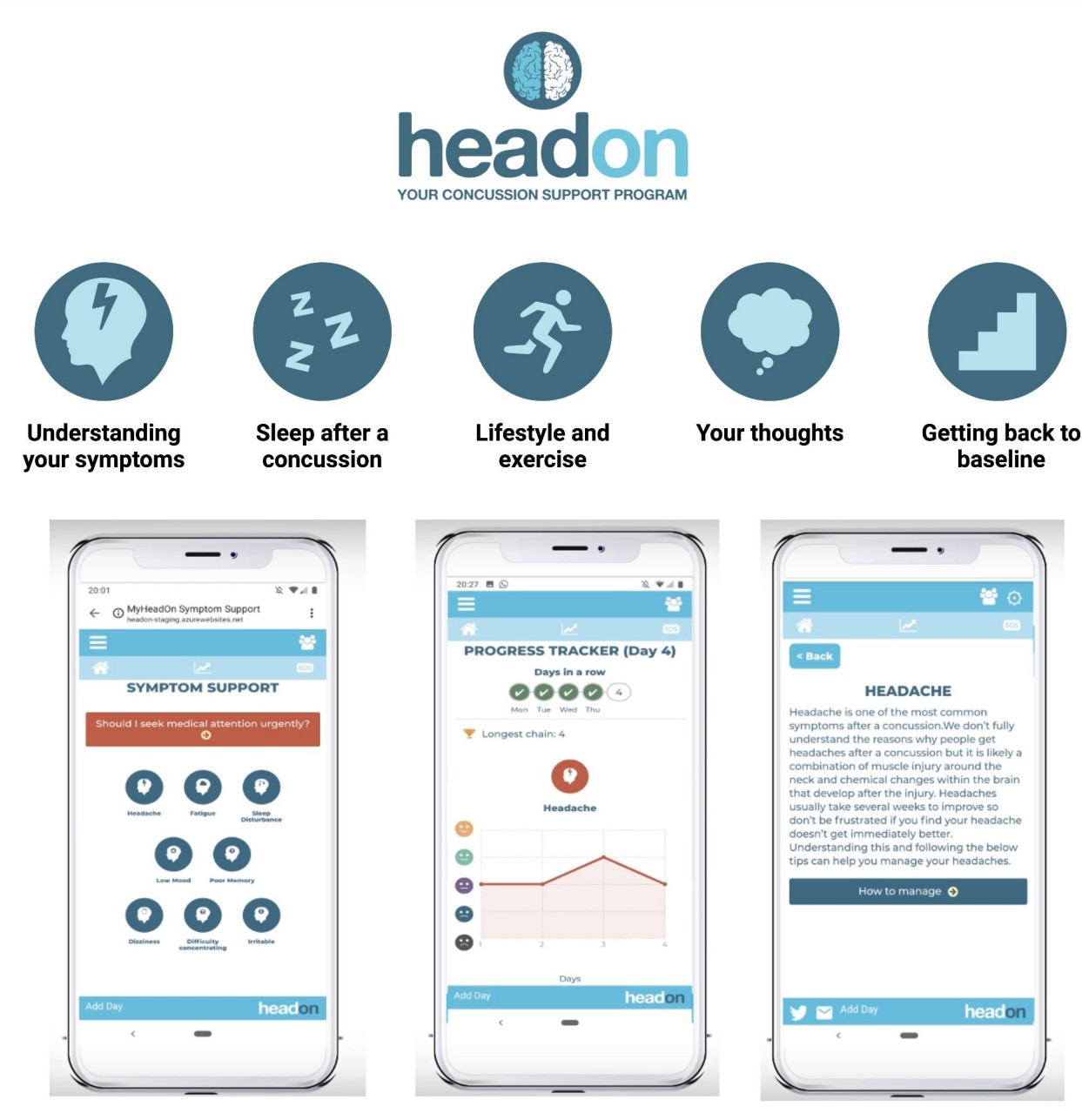

**Fig 1. Demonstration of the HeadOn user interface including the symptom support area and progress tracker.** Image used with permission from HeadOn Health Ltd under the Creative Commons Attribution 4.0 International (CC BY 4.0) license.

approach and the Medical Research Council (MRC) guidance on the development of complex interventions [9,10]. HeadOn is delivered as a web application and runs over 5 stages sequentially, with each stage lasting 7 days (total duration of 35 days). On registering, patients filled out a simple questionnaire covering demographics and whether the concussion was sports-related and if they had consumed alcohol at the time of the injury. They also completed baseline questionnaires including Rivermead Post-Concussion Symptom Questionnaire, Patient Health Questionnaire-9 (PHQ-9) and the FAST alcohol screening test [11–13]. These

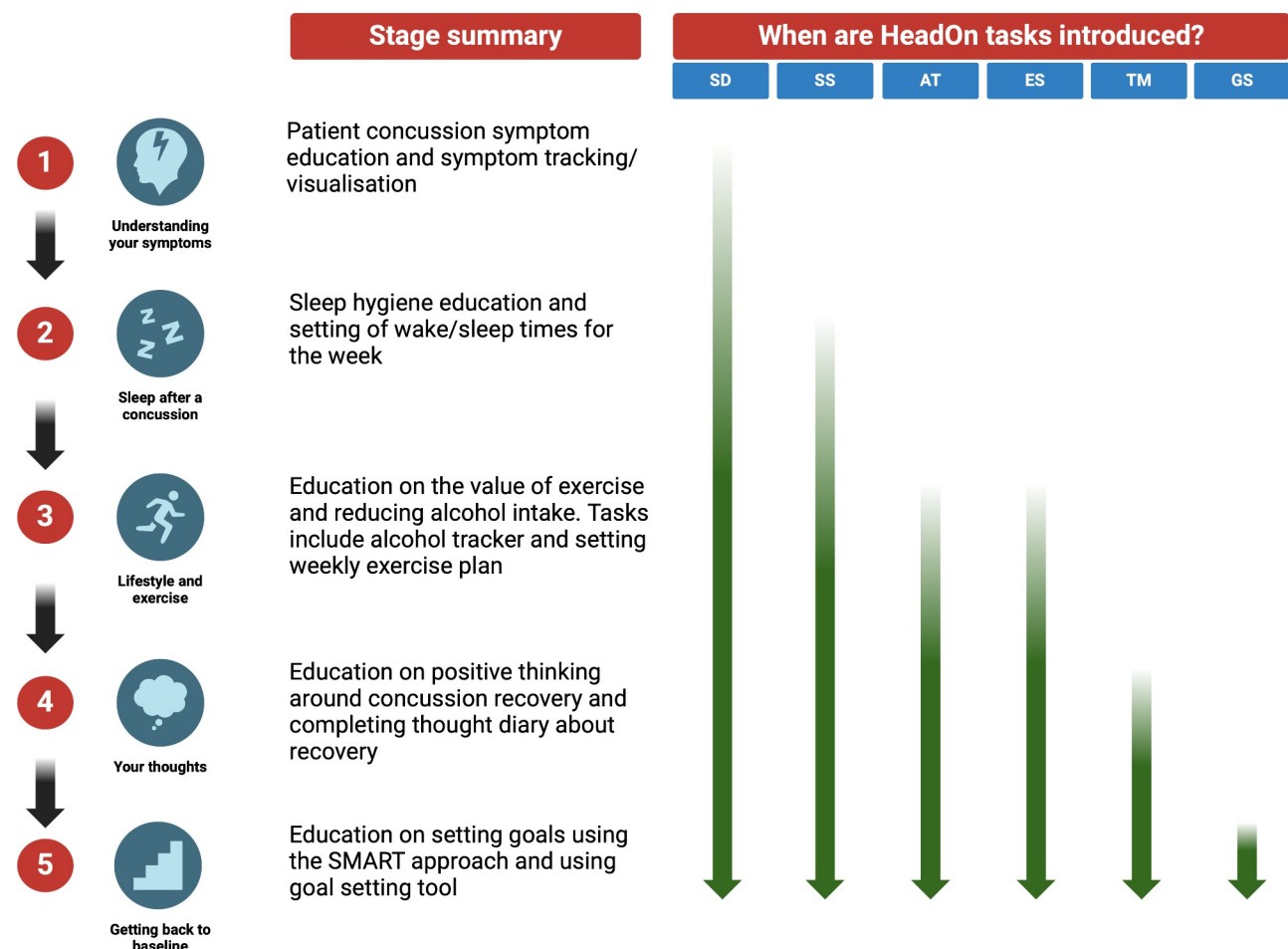

**Fig 2. Schematic of the HeadOn user journey across the sequential 5 stages which each last 1 week, alongside when the core HeadOn tasks are introduced to the user.** SD: Symptom diary; SS: Sleep setting; AT: Alcohol tracker; ES: Exercise setting; TM: Thought monitor; GS: Goal setting. Image used with permission from HeadOn Health Ltd under the Creative Commons Attribution 4.0 International (CC BY 4.0) license.

measures were chosen as they are well validated questionnaires that have been used extensively in concussion research [14,15]. The five stages of HeadOn are (1) Understanding your symptoms, (2) Sleep after a concussion, (3) Lifestyle and exercise, (4) Your thoughts, and (5) Getting back to baseline (**Fig 2**). During each stage, the patient is invited to complete a task and has access to a range of multimedia educational material. The weekly tasks vary by stage and comprise: completing symptom diaries, setting exercise goals, counting alcohol unit consumption, completing a thought diary and setting up a sleep time routine (**Table 1**). Completing the symptom diary is a core task of HeadOn and patients are invited to complete it every day of the 35-day program. The symptom diary can be completed once at any time during the day. This single daily Ecological Momentary Assessment (EMA) data capture at a random time is a well-established sampling methodology [16]. The 35-day cut off was decided in conjunction with patients during the co-development process [9]. The rationale behind it was based on evidence that patient engagement with digital interventions diminishes as patients recover and that the majority of patients' post-concussion symptom burden improves around a month post-injury [17,18]. In the symptom diary, patients can rate 8 post-concussion symptoms from 0 ('none') to 4 ('severe'). The symptoms include: sleep disturbance, dizziness, headache, difficulty concentrating, poor memory, low mood, irritability, and fatigue. At the end of HeadOn,

**Table 1. Summary of the HeadOn digital health intervention, these are consecutive weeks with associated tasks[9].** PHQ-9: Patient Health Questionnaire 9; CBT: Cognitive Behavioural Therapy; SMART: Specific, Measurable, Achievable, Relevant, and Time-Bound.

| Weeks | Intervention component | Description | Task |
|---|---|---|---|
| | Introduction | An introductory video explains what HeadOn involves, and the patient is then invited to complete two questionnaires (Rivermead Post-Concussion Symptom Questionnaire and PHQ-9). | Complete Rivermead questionnaire and PHQ-9 (one-off task) |
| Week 1 | Understanding your symptoms | Stage 1 focuses on providing patients with information about post-concussion symptoms and techniques for managing them. The patient is invited to complete a daily symptom diary and can view their data in the Progress Tracker. | Complete symptom diary (recurring daily task) |
| Week 2 | Sleep after a concussion | Stage 2 focuses on sleep disturbance after a concussion. The audio introduction provides the patient with information on good sleep hygiene. The patient is also invited to set up notifications for a wake-up time and bedtime for the week. | Set wake-up and sleep times (one-off task) |
| Week 3 | Lifestyle and exercise | Stage 3 focuses on 2 areas: physical activity and examining alcohol consumption. All patients are invited to set 3 days of the week to perform non-contact physical activity. If the patient had consumed alcohol at the time of injury, they are invited to set 3 alcohol-free days and use an alcohol tracker to monitor their alcohol consumption for the week. | Set exercise days (one-off task); set alcohol-free days and use alcohol tracker (one-off tasks) |
| Week 4 | Your thoughts | Stage 4 focuses on patient's thoughts about their concussion recovery. The audio introduction discusses CBT concepts regarding the role of thoughts on behavior and emotion. The patient is invited to use a thought diary to examine their thoughts about their recovery. | Complete the thought monitor (one-off task) |
| Week 5 | Getting back to baseline | Stage 5 focuses on supporting the patient to return to their preinjury function. During this week, the patient is invited to set a goal to complete by the end of the program. They are encouraged to use the SMART approach. | Set goal using HeadOn goal setter (one-off task) |

patients were invited to complete the Rivermead and PHQ-9 questionnaires again. A 6-week functional outcome, the Glasgow Outcome Score Extended, was also collected by telephone for a group of the study participants.

## Study participants

Participants were recruited over two study phases. The first phase was a prospective cohort study conducted in the Royal Infirmary of Edinburgh Emergency Department (ED) which ran between November 2021 and April 2022. Inclusion criteria for the study were aged $\geq$16 years presenting with a concussion, which was defined according to the American Congress of Rehabilitation Medicine—a traumatically induced disruption of brain function presenting as any alteration of mental status, loss of consciousness, or posttraumatic amnesia [19]. Patients (n = 50) presenting to the ED with a concussion were invited to participate and use HeadOn freely. After completion of the study, HeadOn was implemented into the Royal Infirmary of Edinburgh ED concussion care pathway and patients diagnosed with a concussion by an emergency physician were signposted to the intervention. A total of 44 patients registered with HeadOn between August and November 2023.

## Statistical analysis

Engagement rates with HeadOn tasks were defined as the percentage of participants who completed a particular task during the course of the HeadOn program (eg: the percentage of participants who completed a thought diary). The normality of continuous data was checked using the Shapiro-Wilk test. The Kruskal Wallis test was used for comparison of non-parametric continuous data and the Chi-squared test for categorical data. For correlational analyses, the Spearman rank correlation coefficient was calculated. For the symptom diary data analysis, the average symptom ratings for each patient were calculated to generate the correlation matrix. K-means clustering was used to categorise the average symptom ratings for each patient across

the 8 post-concussion symptoms. To determine the optimal number of clusters, we applied the Elbow Method examining the inertia decrease across 1 to 6 potential clusters. Following this, K-means clustering was performed with the chosen number of clusters. The results were then visualized using Principal Component Analysis (PCA) to reduce the dataset to two dimensions for easy graphical representation. Data analysis and visualisation were conducted with Python using the Data Analyst function of ChatGPT.

### Ethics statement

The prospective cohort study adhered to the relevant ethical guidelines and regulations, gaining approval from the North West—Preston Research Ethics Committee (reference 21/NW/0211) on September 14, 2021. Participants in this study (n = 50) provided written informed consent prior to involvement. They received no compensation for taking part. After implementation of HeadOn at the Royal Infirmary of Edinburgh ED, patients (n = 44) did not provide consent as HeadOn was part of routine clinical practice. This phase of the study was approved as a service evaluation by the Royal Infirmary of Edinburgh Clinical Governance department. Data collected during the research were securely maintained according to the privacy and data protection policies of the University of Edinburgh and NHS Lothian. The databases utilized met industry standards for safeguarding sensitive information. Data exported from HeadOn underwent de-identification and encryption before analysis.

## Results

### Baseline patient parameters and intervention engagement

A total of 94 patients were included in the study. Average patient age was 41($\pm$16) and 65.9% were female. At the time of concussion, 31 (32.9%) had consumed alcohol and 18 (19.1%) concussions were sports related. The average time from concussion to registration with HeadOn was 7 ($\pm$8) days. Across the six core HeadOn tasks, engagement was as follows: symptoms diary (n = 84, 89.3%), sleep setting (n = 50, 53.1%), exercise planning (n = 39, 41.4%), alcohol tracker (n = 44, 44.6%), thought diary (n = 30, 31.9%) and goal setting (n = 22, 23.4%). Twenty-one (22.3%) participants were high engagers, which was defined as participants completing all the tasks across the 5 stages of HeadOn. **Table 2** summarises patient parameters and user engagement with each of the tasks.

### Post-concussion symptom burden and dynamics

At registration, the average Rivermead was 32 ($\pm$13) and PHQ-9 was 13 ($\pm$7). There was a positive Spearman's rank correlation coefficient between Rivermead and PHQ-9 (0.78; $p < 0.001$). During the study period, a total of 758 symptom diaries were completed by 84 patients (average of 9 diaries per patient). This equated to 6064 individual post-concussion symptom ratings. Overall aggregate symptom scores burdens (total score across 8 symptoms = 5726) revealed the following burden of post-concussion symptomatology: fatigue (864; 15.1%), sleep disturbance (824; 14.4%), difficulty concentrating (724; 12.6%), low mood (710; 12.4%), poor memory (699; 12.2%), headache (695; 12.1%), dizziness (606; 10.6%) and irritability (604; 10.6%). Aggregate daily symptom scores demonstrated wide variation but an overall trend to declining symptom burden over the 35-day HeadOn program (**Fig 3A**). Temporal analysis of individual post-concussion symptoms revealed inter-symptom variation in the recovery dynamics (**Fig 3B**). Physical (headache and dizziness) and emotional (irritability and low mood) symptoms demonstrated earlier rates of recovery from day 14 days of HeadOn. In contrast, it was not until day 28 that improvements in fatigue and sleep disturbance were observed. In the

**Table 2. Summary of patient demographics, baseline patient reported outcome scores and user engagement with the HeadOn tasks across the 5 stages.**

| Patient parameter | Value (SD or %) |
|---|---|
| Age | 41 (±16) |
| Female gender | 62 (65.9%) |
| Alcohol consumption at time of concussion | 31 (32.9%) |
| Sports-related concussion | 18 (19.1%) |
| Time from concussion to HeadOn registration (days) | 7 (±8) |
| Baseline patient reported measures | Average value (SD) |
| Rivermead post-concussion questionnaire | 32 (±13) |
| PHQ-9 | 13 (±7) |
| FAST alcohol score | 3 (±2) |
| HeadOn Engagement rates | Count (%) |
| Symptom diary | 84 (89.3) |
| Setting sleep wake up and bedtime | 50 (53.1) |
| Setting alcohol free days | 44 (46.8) |
| Setting exercise days | 39 (41.4) |
| Thought diary | 30 (31.9) |
| Goal setter | 22 (23.4) |

correlation matrix, there were strong positive correlations between low mood and irritability (r = 0.84), poor memory and difficulty concentrating (r = 0.83), and difficulty concentrating and fatigue (r = 0.79) (**Fig 4**).

## Post-concussion symptom k-means clustering

Next, we applied K-means clustering to categorise the 84 patients who inputted symptom data. We opted for 3 clusters after the application of the Elbow Method (**Fig 5A**). Following this, K-means clustering was performed with the chosen number of clusters. The results were then visualised using Principal Component Analysis (PCA) to reduce the dataset to two dimensions for easy graphical representation (**Fig 5B**). The symptom profiles of each cluster were defined in radar plots (**Fig 5C**). Cluster 0 (n = 24) had a low symptom burden profile across all the post-concussion symptoms. While Cluster 1 (n = 35) had moderate symptom burden but with more pronounced fatigue. Cluster 2 (n = 25) had a high symptom burden profile across all the post-concussion symptoms. Comparison of the clusters demographics and baseline features found no significant difference in the average age or time to registration for HeadOn from injury (**Table 3**). There was a higher percentage of males in Cluster 0 compared to the other two clusters. Baseline Rivermead and PHQ-9 scores were significantly different across the three clusters and reflected the symptom burdens of the clusters throughout HeadOn. At the completion of HeadOn, 28 (29.8%) participants completed the Rivermead and PHQ-9 questionnaires. There was a significant relationship between the symptom clusters for both the Rivermead (p = 0.05) and PHQ-9 (p = 0.003), reflecting the severity of the clusters (**Table 4**). For participants with 6-week functional outcome (n = 27; 28.7%), there was a higher percentage of participants with a GOSE <8 in Cluster 2 (62.5%) compared to Clusters 0 (26.3%) and 1 (37.5%), but this did not reach statistical significance (p = 0.47).

## Qualitative analysis of patient perspectives, concerns, and goals

A total of 53 thought diaries were completed by 30 patients with an average of 1.8 diaries per patient (range 1–7). Thematic analysis demonstrated 7 distinct themes to patient thoughts

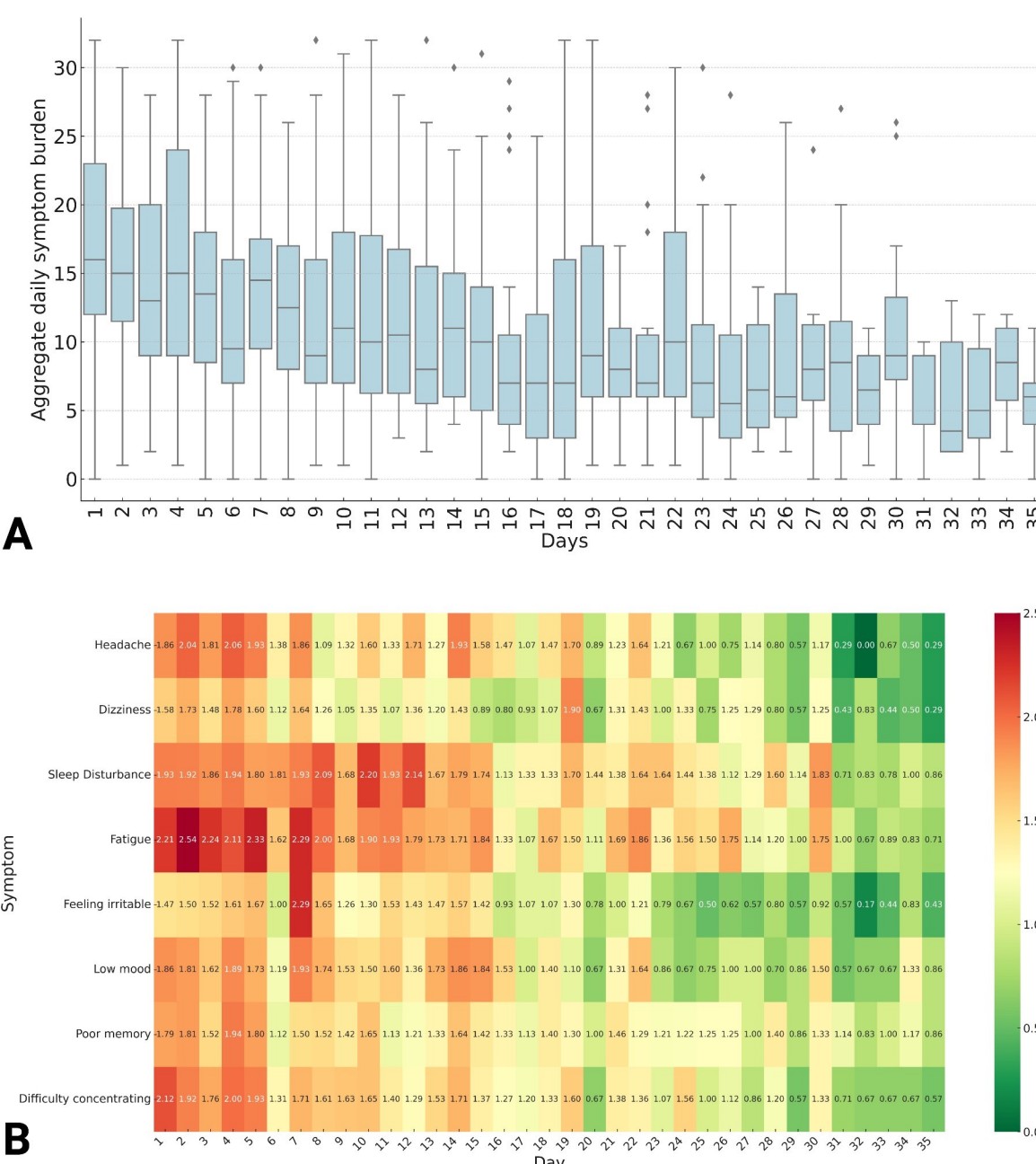

**Fig 3.** Aggregate daily symptom burden over the 35-day HeadOn program (A) heat map depicting temporal changes in individual post-concussion symptom burden over the course of HeadOn (B).

about their concussion recovery. For just under a fifth of thought diary entries (16.9%), patients shared positive feelings and optimism about their recovery. This included entries such as '*Not my fault. Feeling fine, and proud my recovery has been so swift*'. Conversely, the rest of the entries conveyed concern including: 'frustration about their post-concussion symptoms' (18.8%)–"*I was annoyed that it was taking so long to go away because it came at the worst possible time*", 'uncertainty on daily life and the future' (15.1%)—"*How long will it take until I make a recovery and return back to college to complete my course*", 'guilt and regret at not preventing

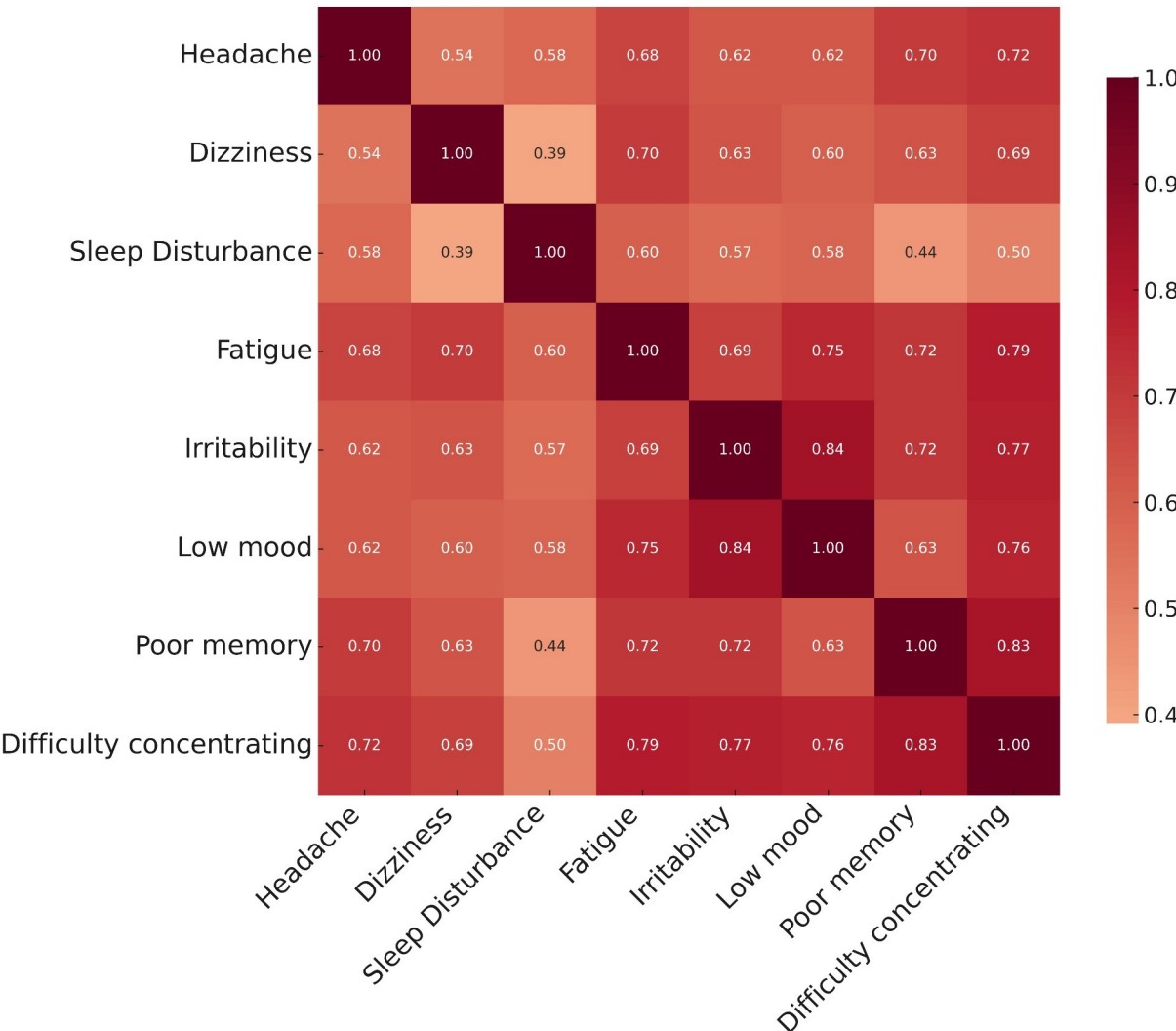

**Fig 4. Correlation matrix of the 8 post-concussion symptoms averaged over the course of HeadOn for 84 patients.**

the incident' (15.1%), 'anxious, lethargic and depressed mood' (13.2%), 'scared about post-concussion symptoms' (11.3%) and 'concern about poor memory and concentration' (9.4%)—*"I feel distressed that I can't remember how I fell and frightened at the sight of blood when I gained consciousness"*. A total of 35 goals were set by 22 patients with an average of 1.6 goals per patient (range 1–6). The most common goal set by patients centred on physical activity (82.9%). This was followed by work-related goals (5.7%), sleep goals (2.9%) and dietary goals (2.9%). The average subjective difficulty rating of achieving these goals (0 = easy, 4 = hard) was 2.3 (±1.1). Just under half (45.7%) of patients recorded in HeadOn that they had attained their goals. The top four reasons quoted by patients on what helped them achieve their goals were: 'being organised' (25.7%), 'being motivated and focused' (25.7%), 'performed coping exercises' (17.1%) and 'getting good sleep' (14.2%). Conversely, the four reasons that patient didn't achieve their goals were: 'being anxious, lethargic and lack of motivation' (50.0%), 'work and lack of time' (17.1%), 'feeling scared and apprehensive' (14.2%) and the 'impact of poor weather' (8.6%).

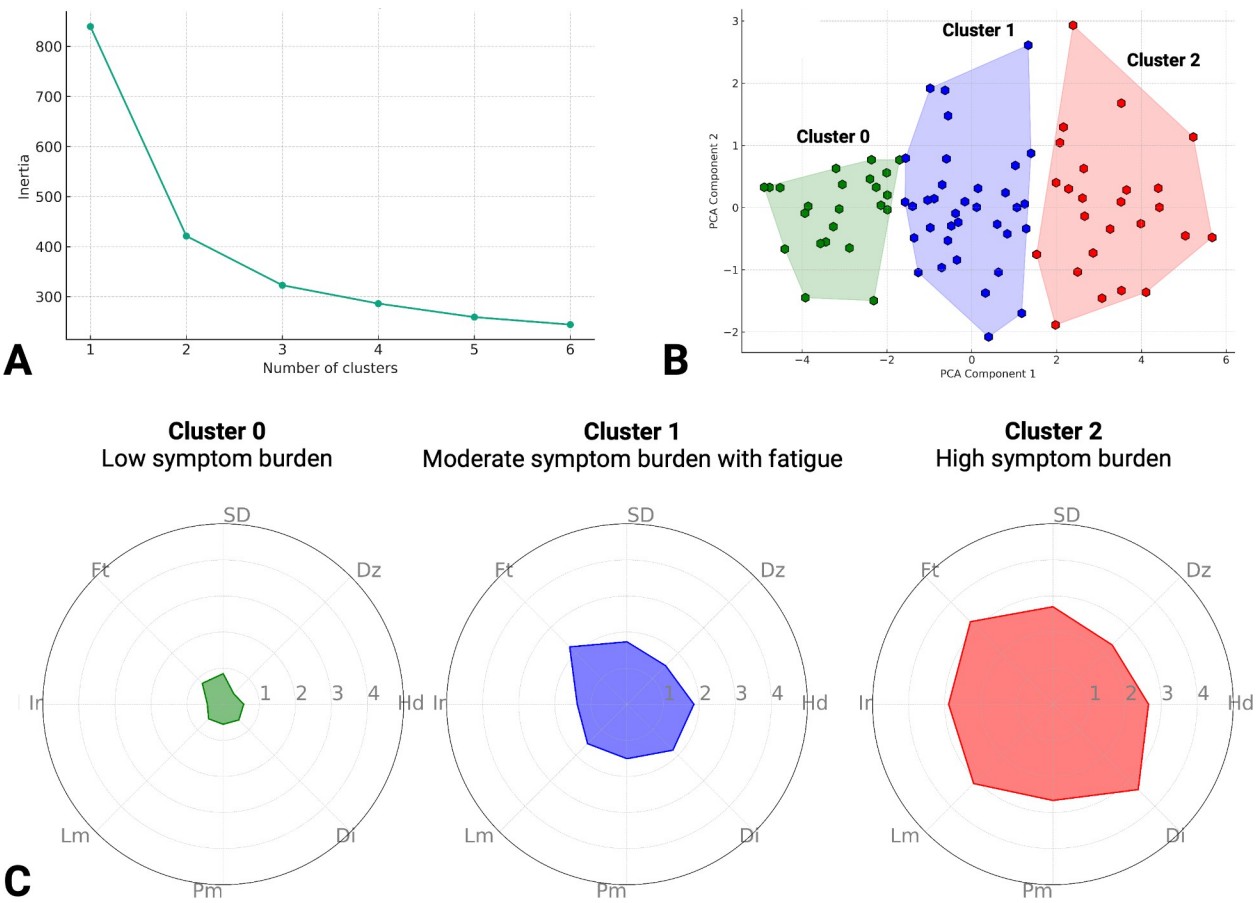

**Fig 5.** Determining the optimum number of clusters using Elbow method (A) k-means clustering visualized using Principal Component Analysis (B) radar plots demonstrating clusters symptom burden. SD: sleeping difficulty; Dz: dizziness; Hd: Headache; Di: difficulty concentrating; Pm: poor memory; Lm: low mood; Ir: irritability; Ft: fatigue.

**Table 3. Comparison of demographics and baseline patient reported scores between the three symptomatology clusters.** PHQ-9: Patient Health Questionnaire 9.

| Data Point | Cluster 0 (n = 24) | Cluster 1 (n = 35) | Cluster 2 (n = 25) | *p* value |
|---|---|---|---|---|
| **Time to registration (days)** | 5.2 (±4.8) | 7.5 (±10.7) | 6.8 (±6.2) | 0.43 |
| **Age** | 45.0 (±16.4) | 40.2 (±15.9) | 35.7 (±14.4) | 0.16 |
| **Baseline Rivermead score** | 20.0 (±9.9) | 35.0 (±9.2) | 41.9 (±8.7) | <0.00001 |
| **Baseline PHQ-9** | 6.9 (±4.7) | 13.1 (±6.1) | 19.1 (±4.8) | <0.00001 |
| **Gender (female)** | 8 (33.3%) | 30 (85.7%) | 19 (76.0%) | 0.00008 |
| **Alcohol consumption at time of injury** | 9 (37.5%) | 10 (28.6%) | 9 (36.0%) | 0.67 |
| **Sport-related concussion** | 5 (20.8%) | 8 (22.9%) | 4 (16.0%) | 0.88 |

**Table 4. Comparison of outcome data at 6 weeks between the three symptomatology clusters.** PHQ-9: Patient Health Questionnaire 9; GOSE: Glasgow Outcome Score Extended.

| Data Point | Cluster 0 | Cluster 1 | Cluster 2 | *p* value |
|---|---|---|---|---|
| **Rivermead (n = 28)** | 8.9 (±7.1) | 16.2 (±9.6) | 40.3 (± 10.0) | 0.005 |
| **PHQ-9 (n = 28)** | 4.7 (±)3.7 | 7.8 (±3.9) | 21.7 (±4.9) | 0.003 |
| **Functional outcome (GOSE <8) (n = 27)** | 4 (26.3%) | 3 (37.5%) | 5 (62.5%) | 0.47 |

## Discussion

Post-concussion symptoms can have a significant impact on patients' function and quality of life [4,7]. Understanding concussion symptomatology and dynamics is vital to inform personalised patient management. In this study, we present one of the largest post-concussion symptomatology datasets which was gathered using digital ecological momentary assessments. This included over 6000 symptom ratings from 84 patients covering 35 days. We identified the most severe post-concussion symptoms experienced by patients using HeadOn. We also confirmed a temporal decline in the burden of symptoms. By applying machine learning approaches, we identified three clusters of patients based on the severity of symptom burden. Ecological momentary assessments have been employed with individuals who have experienced a concussion in the past. Lewandowski and colleagues examined six adolescents (three of whom suffered a concussion and three non-injured controls) using ecological momentary assessments for 5 days [20]. More recently, a further study captured daily symptom data on 34 adolescents who had experienced a concussion for 20 days [21]. A similar study examined 10 adolescents with a recent concussion over 20 days [22]. Our study surpasses what was previously reported in the literature both in terms of the scale and volume of data.

In our sample, we found that fatigue, low mood and difficulty concentrating were the top three symptoms in terms of severity. This is in keeping with previous reports including a longitudinal population study spanning two weeks to 1-year post-injury [23]. The authors found fatigue to be the most prevalent post-concussion symptom. Other studies have suggested that somatic symptoms (headache and dizziness) are prominent in the early stage (1–2 weeks) and then psychosocial symptoms (irritability and low mood) are reported increasingly often in the later period of recovery (4–8 weeks) [24]. We did not find this in our studied cohort who registered for HeadOn an average of 7 days from injury. The temporal profile of symptoms pointed toward a general improvement across all symptom types and no evidence of delayed worsening of psychosocial symptomatology. The trend of improving symptom burden over time has been reported by other authors. Wiebe and colleagues captured ecological momentary assessments over a period of 20 days and found a gradual decline in the severity of post-concussion symptom burden [21]. The same group conducted a larger study examining time to symptom resolution in a cohort of 118 of adolescent patients with sports-related concussion [25]. They found that by using mobile ecological momentary assessments, they could identify patients with symptom resolution between 9–10 days post injury compared to clinical clearance at a clinical consultation which took an average of 14 days. The authors argue that digital ecological momentary assessments allow remote monitoring and expedite treatment decisions.

By using k-means cluster analysis, we identified three clusters of patient post-concussion symptomatology. This machine learning approach has been used to identify symptom-based patient clusters in a variety of diseases including COVID, cervical myelopathy and cancer [26–28]. K-means cluster analysis doesn't demand significant computational power and performs well with moderate datasets of continuous numerical data. This makes it a powerful analytical approach to examining symptom rating data. The symptom severity clusters we identified have the potential to support clinical decision-making and the delivery of targeted support to those in higher risk clusters. However, prior to translation into clinical practice, these clusters need to be both internally validated in a larger dataset from HeadOn (which is on-going) and undergo external validation from another concussion symptom dataset. A recent review of concussion subtypes identified five groups based on symptom profile rather than severity: cognitive, oculomotor, headache-migraine, vestibular and anxiety-mood [2]. These profiles were based on patient assessments within 3 days of concussive injury, earlier than the profiles determined in the current study. The headache-migraine subtype was the most prevalent in

children whereas the cognitive subtype was most prevalent in adults. Interestingly, in our analysis there was evidence of positive correlation between post-concussion symptom subtypes including the cognitive (difficulty concentrating and poor memory) and emotional (irritability and low mood) symptoms. However, when k-means clustering was applied, patients were grouped based on symptom severity rather than symptom-type profile. K-means clustering groups cases based on data proximity in a feature space, which in the context of this study, was defined by symptom ratings. These findings suggest that in our dataset, the variation in symptom severity across cases was more pronounced than the variation in the types of symptoms experienced. This fits with another k-means cluster analysis of concussion patient who identified five clusters where symptom severity was the main differentiator of the cluster phenotypes [29]. A series of outcome measures including the Patient-Reported Outcomes Measurement Information System, Dizziness Handicap Inventory, Pain Catastrophizing Scale, and Immediate Post-Concussion Assessment and Cognitive Testing Tool were analysed retrospectively from 275 patients. This analysis supports our finding that post-concussion symptom heterogeneity appears to lie more with variation in severity of symptom burden, rather than symptom-specific profiles. This holds relevance to determining clinic outcome as there is growing evidence that symptom burden at 2–3 weeks post-injury is a strong predictor of functional outcome at 6 months [30].

There is a fine balance in developing digital interventions for concussion between the value in supporting patients against concerns on the impact of screen time on recovery. Concussion guidance is to limit screen time early after an injury. A clinical trial looking at this question found that patients who abstained from screen time for the first 48 hours after their injury had a shorter recovery than those permitted to engage in screen time [31] In the trial, the screen time–abstinent group had 130 minutes of screen time in the first 3 days after their injury. Although HeadOn is delivered mostly through a screen (mobile or desktop), there is the option of consuming some of the content through audio. Also, HeadOn was designed to be delivered with limited screen time (5–10 minutes per day), which could easily be achieved within the 130 minutes of the intervention group quoted in the screen time RCT.

This study has several limitations. Firstly, patients volunteered to use HeadOn and input data into the symptom diaries. This may have introduced selection bias based both in terms of the participants but also the amount and type of data collected. Alongside this, the fact that the average sign-up time to HeadOn was 7 days means that the data presented in the article likely represents a higher severity group who opted to access support later in their recovery journey. Therefore, caution should be used extrapolating these findings to all de-novo patients at the start of their recovery following a concussion. HeadOn measured symptoms once daily at a random time and therefore it was not possible to provide insights into diurnal variation of concussive symptoms. A once daily approach is not uncommon in brain injury EMA research as it can provide a longer view of symptom burden over weeks or months [32]. For the correlation matrix and k-means clustering, symptom burdens for each patient were averaged across 35-day program meaning these analyses provided an overview of the study period rather than a temporal analysis.

## Conclusions

Post-concussion symptomatology is highly heterogenous and can have a significant impact on patient outcome. By leveraging digital ecological momentary assessments, we have captured a rich dataset of post-concussion symptom burden over a 35-day period. K-means analysis identified three patient clusters based on the severity of symptom burden. By leveraging these approaches, patients with high symptom burden can be identified and supported at scale to meet the public health challenge of concussion.

## Acknowledgments

We would like to extend our gratitude to all the patients who participated in this study and took the time to use HeadOn. We would also like to thank Julia Grahamslaw and Alison Grant from the Emergency Medicine Research Group Edinburgh (EMERGE) who supported the delivery of the research study. Finally, we would like to thank Dr Molly Brewster for her work on implementing HeadOn into the concussion care pathway at the emergency department in The Royal Infirmary of Edinburgh. Finally, we would like thank our funders: The NIHR Brain Injury MedTech Co-operative and the Scottish EDGE.

## Author Contributions

**Conceptualization:** Rebecca Blundell, Christine d'Offay, Alan Carson, David Gillespie, Matthew Reed, Aimun A. B. Jamjoom.

**Data curation:** Christine d'Offay, Aimun A. B. Jamjoom.

**Formal analysis:** Rebecca Blundell, Aimun A. B. Jamjoom.

**Funding acquisition:** Christine d'Offay, Charles Hand, Alan Carson, David Gillespie, Matthew Reed, Aimun A. B. Jamjoom.

**Investigation:** Charles Hand, Daniel Tadmor, Aimun A. B. Jamjoom.

**Methodology:** Rebecca Blundell, Christine d'Offay, Charles Hand, Daniel Tadmor, Alan Carson, David Gillespie, Matthew Reed, Aimun A. B. Jamjoom.

**Project administration:** Christine d'Offay, Charles Hand, Daniel Tadmor, Alan Carson, David Gillespie, Matthew Reed, Aimun A. B. Jamjoom.

**Resources:** Aimun A. B. Jamjoom.

**Supervision:** Alan Carson, David Gillespie, Matthew Reed, Aimun A. B. Jamjoom.

**Validation:** Rebecca Blundell, Christine d'Offay, Aimun A. B. Jamjoom.

**Visualization:** Aimun A. B. Jamjoom.

**Writing – original draft:** Rebecca Blundell, Aimun A. B. Jamjoom.

**Writing – review & editing:** Rebecca Blundell, Christine d'Offay, Charles Hand, Daniel Tadmor, Alan Carson, David Gillespie, Matthew Reed, Aimun A. B. Jamjoom.

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
