## [Decision Letter · Decision Letter 0]

29 Jul 2024

PDIG-D-24-00173

Post-concussion symptom burden and dynamics: insights from a digital health intervention and machine learning

PLOS Digital Health

Dear Dr. Jamjoom,

Thank you for submitting your manuscript to PLOS Digital Health. After careful consideration, we feel that it has merit but does not fully meet PLOS Digital Health's publication criteria as it currently stands. Therefore, we invite you to submit a revised version of the manuscript that addresses the points raised during the review process.

Please submit your revised manuscript within 60 days Sep 27 2024 11:59PM. If you will need more time than this to complete your revisions, please reply to this message or contact the journal office at digitalhealth@plos.org. Please include the following items when submitting your revised manuscript:

We look forward to receiving your revised manuscript.

Kind regards,

Dhiya Al-Jumeily OBE, PhD

Section Editor

PLOS Digital Health

Journal Requirements:

1. Please send a completed 'Competing Interests' statement, including any COIs declared by your co-authors. If you have no competing interests to declare, please state "The authors have declared that no competing interests exist". Otherwise please declare all competing interests beginning with the statement "I have read the journal's policy and the authors of this manuscript have the following competing interests:"

2. Please provide separate figure files in .tif or .eps format.

Additional Editor Comments (if provided):

Reviewers' comments:

Reviewer #1: Thank you for this work. Digital health interventions are an an area of concussion management that are often underexplored. To that end I am glad to see this project and its success. I commend the authors on embedding this tool into the concussion pathway for the second part of the study. This is often a challenging task both from an administrative and clinical policy standpoint.

I have some comments and points of clarification which would strengthen this work:

The development of the digital health tool is well described. Given that HeadOn is a web application, can the authors comment on the use of screens during this cognitive rest period post-concussion? How might this affect recovery? I note that a brief registration as well as Rivermead and PHQ are done in the initial phases. These are relatively short questionnaires, but has consideration been given to whether using a web application to complete clinical questions, document the symptom diary, and review educational materials on a daily basis soon after concussion increases cognitive overload and mental fatigue in these patients ?

There is some evidence supporting the recommendation that patients abstain from screen time in the first 48-72 hours following a concussion to decrease time to symptom resolution, thus shortening the timeline to return to their usual daily activities.

Table 1: are the stages concurrent or consecutive? 

Do you use “symptomology” and “symptomatology” synonymously?

How was the 35 day cut off chosen for HeadOn intervention?

Statistical analysis: 1. I am curious why the data analyst function of ChatGPT was used for analysis when Python can perform the modeling and visualizations.

2. Provide the rationale for selecting k means over other clustering algorithms.

3. Provide some comment on the statistical power of the study sample for clustering analysis.

4. How was missing data handled?

In terms of the causes, I note that there is a question for whether this was a sports-related concussion (19%)- what were the main mechanisms? 

The range of days from injury to registration was about one week. What do you think is the effect of this variability on the severity of reported symptoms?

Please provide some context for using Rivermead, PHQ-9, and FAST.

Define “engagement rate” in Table 2.

Figures: I strongly encourage you to make the scientific figures accessible to readers with color-blindness. Provide a legend for abbreviations used in the radar plots.

Discussion: the sentence “These clusters can be integrated into individuals’ current

297 electronic health record systems to assist in clinical decision-making” needs to be qualified. Is external validation using another dataset or further data from ongoing recruitment at your site required before using the clusters in clinical practice?

Reviewer #2: Thank you for this work. Digital health interventions are an area of concussion management that are often underexplored. To that end I am glad to see this project and its success. I commend the authors on embedding this tool into the concussion pathway for the second part of the study. This is often a challenging task both from an administrative and clinical policy standpoint.

I have some comments and points of clarification which would strengthen this work:

The development of the digital health tool is well described. Given that HeadOn is a web application, can the authors comment on the use of screens during this cognitive rest period post-concussion? How might this affect recovery? I note that a brief registration as well as Rivermead and PHQ are done in the initial phases. These are relatively short questionnaires, but has consideration been given to whether using a web application to complete clinical questions, document the symptom diary, and review educational materials on a daily basis soon after concussion increases cognitive overload and mental fatigue in these patients ?

There is some evidence supporting the recommendation that patients abstain from screen time in the first 48-72 hours following a concussion to decrease time to symptom resolution, thus shortening the timeline to return to their usual daily activities.

Table 1: are the stages concurrent or consecutive? 

Do you use “symptomology” and “symptomatology” synonymously?

How was the 35 day cut off chosen for HeadOn intervention?

Statistical analysis: 1. I am curious why the data analyst function of ChatGPT was used for analysis when Python can perform the modeling and visualizations.

2. Provide the rationale for selecting k means over other clustering algorithms.

3. Provide some comment on the statistical power of the study sample for clustering analysis.

4. How was missing data handled?

In terms of the causes, I note that there is a question for whether this was a sports-related concussion (19%)- what were the main mechanisms? 

The range of days from injury to registration was about one week. What do you think is the effect of this variability on the severity of reported symptoms?

Please provide some context for using Rivermead, PHQ-9, and FAST.

Define “engagement rate” in Table 2.

Figures: I strongly encourage you to make the scientific figures accessible to readers with color-blindness. Provide a legend for abbreviations used in the radar plots.

Discussion: the sentence “These clusters can be integrated into individuals’ current

297 electronic health record systems to assist in clinical decision-making” needs to be qualified. Is external validation using another dataset or further data from ongoing recruitment at your site required before using the clusters in clinical practice?

Reviewer #3: A bit more information about the HeadOn web-based tool would have been useful. This is likely a symptom diary with some educational resources. The daily diary is used to collect data on symptoms from the patients every day for 35 days. Does the HeadOn system require that the patient complete symptom diary at various points in time or just once a day (and at what time)? It is also not clear if the time at which the symptoms were entered were considered during the data analysis. The strength of ecological momentary assessment (EMA) technique is to access patient at multiple times and as they go by their daily activities (e.g., exercise, being tired, resting etc.), and it doesn’t seem that way. In order to inform personalized management for concussion patients we need better understanding of the dynamic fluctuations in the post-concussion symptoms within patients as well as between patients. Can the author comment on how symptom collection via HeadOn account for the diurnal variations and variations related to daily activities? In particular how lack of this information could have impacted the results? Also, whether once-a-day symptom diary could be truly regarded as EMA? 

One of the main symptoms that patients with concussion include Cognitive symptoms (e.g., difficulty concentrating and memory problems). Can the author comments how this was considered while designing headOn, given that it requires a lot of data entering and viewing of educational resources. There are many sensors and wearable, and voice activated approaches towards EMA. Why were they not used? 

On line 123, “The weekly tasks vary by stage and compromise”. I think you mean to say comprise.

Regarding five stages of HeadOn:(1) Understanding your symptoms, (2) Sleep after a concussion, (3) Lifestyle and exercise, (4) Your thoughts, and (5) Getting back to baseline. Are these to be completed sequentially? The process is not clear. Perhaps authors might draw a diagram showing the dependency among the stages. 

The questionnaires Rivermead, PHQ-9, and FAST were used as baseline patient reported measures. Some rationale for their usage must be provided. 

Why a timeframe of 35 days chosen for the data collection? Is this the usual time by which the concussion symptoms eventually resolved in most patients? Perhaps some context with some evidence could be included. 

The studied cohort registered for HeadOn an average of 7 days from injury. What was the impact of this time gap on symptom severity. 

Line 163: “Data analysis and visualisation were conducted with Python using the Data Analyst function of ChatGPT: I am not clear what does this mean? Python was used for data analysis, so what was the purpose of Data Analyst function of ChatGPT?

Did the authors consider clustering algorithms other than k-means for clustering? If not, why not. What is the rationale for using k-means for this data? Are there any particular strengths of k-means with regards to this particular problem and the data set?

Some information on data preprocessing steps must be provided. E.g., how clean this data was, how was the missing or erroneous data handled. This data is self-reported, as well as time-series, were any specific methodological decision, e.g., data preprocessing strategies utilized with regards to these issues.

---

## [Decision Letter · Decision Letter 1]

10 Nov 2024

Post-concussion symptom burden and dynamics: insights from a digital health intervention and machine learning

PDIG-D-24-00173R1

Dear Mr. Jamjoom,

We are pleased to inform you that your manuscript 'Post-concussion symptom burden and dynamics: insights from a digital health intervention and machine learning' has been provisionally accepted for publication in PLOS Digital Health.

Best regards,

Dhiya Al-Jumeily OBE, PhD

Section Editor

PLOS Digital Health

**Comments to the Author**

Reviewer #1: All comments have been addressed

Reviewer #3: All comments have been addressed
